# Differential Transcription of SOCS5 and SOCS7 in Multiple Sclerosis Patients Treated with Interferon Beta or Glatiramer Acetate

**DOI:** 10.3390/ijms21010218

**Published:** 2019-12-28

**Authors:** Emmanuel Rojas-Morales, Gerardo Santos-López, Samuel Hernández-Cabañas, Raúl Arcega-Revilla, Nora Rosas-Murrieta, Carolina Jasso-Miranda, Elie Girgis El-Kassis, Julio Reyes-Leyva, Virginia Sedeño-Monge

**Affiliations:** 1Decanato de Ciencias de la Salud, Facultad de Medicina, Universidad Popular Autónoma del Estado de Puebla, Puebla 72410, Mexico; emmanuel.rojasm01@gmail.com; 2Laboratorio de Biología Molecular y Virología, Centro de Investigación Biomédica de Oriente, Instituto Mexicano del Seguro Social, Metepec, Puebla 74360, Mexicojulio.reyes@imss.gob.mx (J.R.-L.); carolinajassom@yahoo.com.mx (C.J.-M.); 3Instituto Mexicano del Seguro Social, Puebla 72560, Mexico; samuel.hernandez01@upaep.edu.mx (S.H.-C.); raularcega@yahoo.com.mx (R.A.-R.); 4Laboratorio de Bioquímica y Biología Molecular, Centro de Química, Instituto de Ciencias, Benemérita Universidad Autónoma de Puebla, Puebla 72570, Mexico; nora.rosas@correo.buap.mx; 5Decanato de Ciencias Biológicas, Universidad Popular Autónoma del Estado de Puebla, Puebla 72410, Mexico; eliegirgis.elkassis@upaep.mx

**Keywords:** multiple sclerosis, JAK-STAT, SOCS5, SOCS7, IFN-*β*, glatiramer acetate

## Abstract

The participation of proinflammatory cytokines in the progression of Multiple Sclerosis (MS) has been well documented. Cytokines activate the JAK-STAT pathway, in which the suppressors of cytokine signaling (SOCS) exert a negative feedback. This paper analyzes the levels of SOCS5 and SOCS7 transcripts, quantified by RT-qPCR, in MS patients, and the concentrations of proinflammatory cytokines, IFN-*γ*, IL17, and IL6, determined by ELISA. Samples of peripheral blood were obtained from MS patients in the relapsing–remitting phase, treated with IFN-*β* or glatiramer acetate (GA), and from healthy individuals. SOCS7 mRNA was significantly higher in patients treated with GA (1.36 ± 0.23) than in those treated with IFN-*β* (0.65 ± 0.1). Regarding gender, the level of SOCS5 and SOCS7 transcripts were similar between MS and healthy females; in MS males, the level of SOCS7 transcripts were significantly lower (0.59 ± 0.03) than in healthy males (1.008 ± 0.05). Plasmatic levels of IFN-*γ* were significantly higher in MS patients (60 pg/mL, range 0–160) than in healthy subjects (0 range, 0–106). The same pattern was observed in MS patients treated with IFN-*β* (68 pg/mL, range 0–160) compared to patients treated with GA (51 pg/mL, range 0–114), and in MS females (64 pg/mL, range 0–161) compared to healthy females (0, range 0–99). We hypothesize that the increase in SOCS7 transcription in patients treated with GA could partially explain the action mechanism of this drug, while the increase in the concentration of IFN-*γ* in MS patients could help elucidate the immunopathology of the disease.

## 1. Introduction

Multiple sclerosis (MS) is an autoimmune, chronic, and degenerative disease of the central nervous system (CNS), characterized by inflammation and demyelination of axonal tissue. Its prevalence has been rising since the last decade, affecting 2.1 million individuals around the world [1]. Women are more affected than men, the ratio being 3:1 [2]. Four forms of MS are known: the classic relapsing–remitting form (RRMS) affects 85% of MS patients; it becomes progressive, a stage defined as secondary progressive (SPMS). Fifteen percent of MS patients can present a progressive form from the onset, named primary progressive (PPMS) [3], while the progressive-relapsing form (PRMS) is found in only a few patients [4]. Over the past two decades, the treatment of MS has changed enormously; more than a dozen drugs are currently approved for RRMS, each with different modes of action. These drugs include disease-modifying drugs (DMDs), considered as first-line treatments, such as interferon-*β* (IFN-*β*) and glatiramer acetate (GA), among others, which interfere with the synthesis of proinflammatory cytokines and the migration of cells into the CNS, reducing the severity and frequency of the disease [5,6]. The etiology of MS is unknown, and the exact molecular mechanisms underlying its onset and progression are still unclear. Both immunity types, i.e., innate and adaptative, contribute to the development and progression of the disease [7]. Autoreactive T cells have been shown to target self-antigens in the CNS [8]. The participation of proinflammatory cytokines, such as IL-4, IL-6, IL-23, IL-27, IL-17, and IFN-*γ*, in the progression of the disease has also been reported. Plasmatic IFN-*γ* levels have been reported to increase in MS patients [9], associated with periods of relapse and a worsening of disease symptoms [10], as well as with fatigue and depression [11]. MS is assumed to be a T-helper 1 (Th1)/Th17-mediated autoimmune disease [12]. An increased expression of IL17 has been correlated both with the severity of MS and with the number of active plaques [13,14,15]. Furthermore, decreased levels of IL17 have been reported in MS patients treated with IFN-*β* compared with non-treated patients [16]. IL-6 is suggested to be the first cytokine in a cascade that participates in an autocrine growth loop leading to Th17 differentiation [17]. In RRMS patients, higher serum concentrations of IL-6 were found compared to healthy controls, with a positive correlation with the number of relapses in female MS patients [18]. Moreover, some correlation between blood IL-6 levels and EDSS has been reported [19]. Most cytokines activate the JAK STAT signaling pathway, which should be finely regulated; among the most important negative regulators is the family of suppressors of cytokine signaling (SOCS), comprising eight members: SOCS1–SOCS7 and CIS. These proteins bind to JAK members and suppress cytokine signaling [20,21,22]. SOCS1, SOCS3, SOCS5, and SOCS7 seem to be involved in the physiopathology of MS. SOCS1 and SOCS3 transcripts have already been analyzed in a previous work, which found higher levels of SOCS1 and lower levels of SOCS3 transcripts [23]. SOCS5 is known to inhibit IL-4 signaling through STAT6 or JAK1, preventing the activation of IL-4 by inhibiting autophosphorylation [24,25]. Similarly, SOCS7 inhibits prolactin signaling by means of STAT3 or STAT5, and it is also associated with the inhibition of IL-6 and IL-23 signaling [26,27]. However, there is scarce information about the relationship between MS and these SOCS members. The aim of the present study was to quantify the levels of SOCS5 and SOCS7 transcripts, as well as the plasmatic concentrations of IFN-*γ*, IL17, and IL16 in samples from MS patients treated with GA or IFN-*β*, and correlate them with the progression and severity of the disease.

## 2. Results

### 2.1. Demographic Characteristics and Clinical Parameters

Thirty-six patients of both genders were included (21 females and 15 males). The mean age was 36.8 ± 8.7 (23–61) years old. The average age of men was 37.4 ± 8.5 (24–55); the average age of women was 38.09 ± 9.6 (23–61).

All patients were in the RR phase of the disease; samples were obtained during the remission phase. The EDSS mean score was 2.6 ± 1.36 for all MS patients, 2.08 ± 0.88 for the group treated with IFN-*β*, and 4.3 ± 1.7 for the group treated with GA. The disease affected mainly three functional systems: sensitive (*n* = 24, 72.7%), pyramidal (*n* = 23, 69.7%) and cerebral cognitive (*n* = 22, 66.7%). The mean time of the evolution of the disease was 6.5 ± 4.1 years.

Twenty-five patients were being treated with 6, 12, or 8 MIU (Millions of International Units) of IFN-*β*, administered once, three times a week and every third day, respectively, (2 patients were treated with IFN-*β* 6 MIU, 8 patients with IFN-*β* 12 MIU, and 15 patients with IFN-*β* 8 MIU). Eleven patients were treated with 20 mg of GA daily. Table 1 shows the main demographic and clinical data, as well as the transcript levels and cytokine concentrations of the patients.

### 2.2. Transcription of SOCS5 and SOCS7 in MS Patients

The relative expression of *SOCS5* and *SOCS7* genes in patients with MS was compared with that of subjects in the control group using the 2^−ΔΔCT ± S^ method.

THE SOCS5 and *SOCS7* transcript levels didn’t show significant differences (*p* = 0.3) between MS patients (0.91 ± 0.14 and 0.82 ± 0.13 respectively) and the control group (1.006 ± 0.16 and 1.008 ± 0.18) (Figure 1A).

With respect to gender, the *SOCS5* and *SOCS7* transcript levels of MS females (0.92 ± 0.05 and 0.9± 0.03 respectively) were similar to those of the control group (1.007 ± 0.06 and 1.008 ± 0.06) (*p* = 0.05). *SOCS7* transcript levels were significantly lower (0.59 ± 0.03, *p* = 0.007) in MS males compared to healthy males (1.008 ± 0.05). Regarding *SOCS5*, no significant reduction was found in MS (0.88 ± 0.04; *p* = 0.2) males compared with healthy males (1.005 ± 0.06) (Figure 1B).

### 2.3. Analysis of SOCS5 and SOCS7 Transcript Levels by Treatment

The transcript levels of SOCS5 in MS patients treated with IFN-*β* or GA were 0.9112 ± 0.14 and 0.9317 ± 0.14 respectively; no significant difference was found between the treatments (*p* = 0.9). With respect to *SOCS7* transcripts, the MS patients treated with GA showed a significant (*p* = 0.03) increase (1.36 ± 0.23) compared with MS patients treated with IFN-*β* (0.65 ± 0.10), which suggests that the treatment with GA increases *SOCS7* transcription (Figure 2A). The data of SOCS7 transcription in MS males treated with GA showed that transcript levels were 1.016 (data no shown), i.e., very close to 1.008, which was the figure for SOCS7 transcript levels in healthy men. This indicated that GA might restore SOCS7 levels in MS males, elevating them to the level of the control group. Thus, MS males treated with GA maintained high SOCS7 transcript levels.

The treatment scheme was as follows: first, IFN-*β* doses of 6 MIU once a week; if the patients showed no improvement, that is, if they suffered new attacks or they became more severe, the treatment was changed to IFN-*β* doses of 12 MIU administered three times a week. Again, if the patients didn’t show any improvement (considering the same criteria), the treatment was changed to IFN-*β* doses of 8 MIU administered every third day. Glatiramer acetate was administered as the last alternative following the same criteria. The transcript levels of *SOCS5* in MS patients receiving one of the three treatments were 0.6242 ± 0.08, 0.9463 ± 0.09 and 0.9370 ± 0.12, respectively, (*p* = 0.5). The transcript levels of SOCS7 were 0.67 ± 0.03, 0.71 ± 0.07 and 0.57 ± 0.07, respectively (Figure 2B). No significant (*p* = 0.2) differences were found in SOCS5 or SOCS7 transcript levels between the different treatments.

The first and third treatments consisted of IFN-*β* 1A (6 and 8 MIU); the second treatment consisted of IFN-*β* 1B (12 MIU). With respect to the type of IFN-*β*, in MS patients treated with IFN-*β* 1A and IFN-*β* 1B, *SOCS5* transcript levels were significantly (*p* = 0.0001, *p* = 0.0003) higher (0.94 ± 0.05 and 0.91 ± 0.11) compared to SOCS7 transcript levels (0.48 ± 0.02) and (0.67 ± 0.02) (Figure 2C). 

### 2.4. Quantification of Cytokines

All patients were analyzed for *SOCS5* and *SOCS7* expression; however, due to the small sampling volume, ILs levels were measured only in 33 MS patients.

The plasmatic concentrations of IFN-*γ*, IL17, and IL6 were measured in MS patients and compared with the control group. ILs are expressed in median and range; upper and lower limits of IFN-*γ* increased significantly (*p* = 0.001) in MS patients treated with GA or IFN-*β* by 60 range (0–160), compared with the control group 0 (0–106) (Figure 3A). With respect to IL17, there was a slight and nonsignificant (*p* = 0.4) increase in MS patients by 1.9 (0–15) and 1.2 (0–15), respectively (Figure 3B), while no significant (*p* = 0.8) difference was found in IL6 (0–18) compared with the control group (0–39) pg/mL, (Figure 3C). IFN-*γ* levels increased compared with other cytokines in both study groups.

The levels of IFN-*γ*, IL17, and IL6 in patients with MS under treatment with IFN-*β* were 68 (0–160), 1.2 (0–14), and 0.3 (0–16) respectively. In patients treated with GA, the levels of IFN-*γ*, IL17, and IL6 were 51 (0–114), 4 (0.2–15), and 0 (0–18) respectively (Figure 3A–C). IFN-*γ* levels were significantly (*p* = 0.003) higher in patients treated with IFN-*β* compared with those treated with GA (Figure 3A).

With respect to the type of IFN-*β* (1A or 1B), the levels of IFN-*γ*, IL17, and IL16 were 38 (14–160), 0.6 (0–7.6), and 0 (0–2) for IFN-*β* 1A; for IFN-*β* 1B, the levels were 91 (0–131), 1.7 (0–13), and 1.4 (0–16) pg/mL (Figure 4A).

With respect to gender, cytokine concentrations (IFN-*γ*, IL17, and IL16) were 53 (0–131), 1.6 (0–15), and 0 (0–18) in MS males and 64 (0–161), 2.2 (0–13), and 1.5 (0–29) pg/mL in MS females, respectively (Figure 4B). A significant increase (*p* = 0.000) of IFN-*γ* concentrations was observed in MS females compared with healthy females 0 (0–99) pg/mL (Figure 4C).

### 2.5. Correlation Analysis

#### 2.5.1. SOCS5/7 Transcription and Disease Activity

Spearman’s correlation coefficient was calculated between *SOCS5* or *SOCS7* transcript levels and disease activity, the latter of which was estimated through the EDSS score. A nonsignificant correlation was found between *SOCS5* and EDSS (*r* = −0.2690, *p* = 0.11) (Figure 5A), while for *SOCS7*, the correlation was significant (*r* = −0.34, *p* = 0.04), suggesting that greater disease activity corresponded to higher *SOCS7* transcript levels (Figure 5B). A significant strong and negative correlation was also found between *SOCS5* transcript levels from patients treated with IFN-*β* and the EDSS score (*r* = −0.6, *p* = 0.0006) (Figure 5C).

#### 2.5.2. SOCS Transcript Levels and Cytokine Concentrations

Spearman’s correlation coefficient was calculated between *SOCS5* or *SOCS7* transcript levels and the concentration of each cytokine. Correlation coefficients lower than 0.2 are not shown. Only a positive and relatively weak correlation was observed between IFN-*γ* concentrations and *SOCS5* transcript levels (*r* = 0.34, *p* = 0.05) in patients treated with GA (Figure 5D).

## 3. Discussion

MS is an autoimmune and neuroinflammatory disease characterized by disabling symptoms, depending of the demyelinating sites affected. The etiology of MS is not completely known. It has been recognized that several proinflammatory cytokines play a role in the physiopathology of MS, activating signaling pathways such as JAK-STAT and some others. The aim of the present work was to determine the transcript levels of SOCS5 and SOCS7 and the plasmatic concentrations of IFN-*γ*, IL17, and IL16 in MS patients in the RR phase. The results showed no significant differences in the transcript levels of *SOCS5* or *SOCS7* between MS patients and the control group, in contrast with other authors who have reported a significant decrease in the expression of *SOCS5* in MS patients [28].

First-line treatment is still based on the use of IFN-*β* and GA. In our study group, 25 and 11 MS patients were treated with IFN-*β* and GA respectively. We found that patients treated with GA showed significantly higher transcript levels of *SOCS7* than patients treated with IFN-*β*. This result is important, because some reports have shown that GA suppresses the activation of STAT1 and STAT3 in rat glial cells, reducing the phosphorylation of those proteins and the activity of IFN-*γ*-activated GAS [29]. Furthermore, it has been reported that *SOCS7* inhibits the IL6 and IFN-*γ* signaling pathways [27]. Thus, an increase in the transcript levels of SOCS7 could inhibit the signaling initiated by proinflammatory cytokines, and may be an important element in the pathogenesis of the disease.

In agreement with this, *SOCS7* has been reported to increase in monocytes derived from MS patients and stimulated with simvastatin, which has immunomodulator effects and promotes the immune response to Th2 cytokines [27]. Simvastatin has been considered as a possible treatment for MS due to its ability to inhibit the formation of CNS lesions and ameliorate the severity of EAE [30,31]. According to our results, GA and simvastatin seem to have similar action mechanisms, i.e., an increase in SOCS7 transcription that leads to an improvement of MS/EAE conditions. Moreover, it is known that GA reduces the number of disease relapses. A study on leukocytes showed that ligands such as prolactin, the human growth hormone, and IL6 activate *SOCS7* [32], which interacts with STAT3 and STAT5, inhibiting their nuclear translocation [26]. It has also been shown that GA decreases the production of IL 2 in T cells cocultured with glial cells [29]. Ye Hyeon Ahn et al. reported that STAT1 and 3 are potential signaling molecules in glial and T-cells associated with neuroinflammation, and also that GA modulates the activation of STAT1 and 3. In the present work, it was found that MS patients treated with GA had a higher EDSS score than patients treated with IFN-β, which could be explained by the treatment scheme, in which GA is indicated as the last option. If treatment with IFN-*β* does not reduce the EDSS score, and if it the clinical condition of the patient does not improve, the patient is treated with GA; thus, patients treated with GA are associated with a longer evolution time of the disease.

In the present study, the measurement of the plasmatic concentrations of proinflammatory cytokines showed a significantly higher level of IFN-*γ* in MS patients compared to healthy individuals. This was expected, since IFN-*γ* is a proinflammatory cytokine whose role in the pathophysiology of MS has been demonstrated [9]. Furthermore, we found significantly lower amounts of IFN-*γ* in patients treated with GA compared with patients treated with IFN-*β*. These results led us to think that IFN-*γ* might be better controlled with GA.

Regarding gender, it is known that MS is more frequent in women than in men [33]. In the present study, we found significantly higher plasmatic concentrations of IFN-*γ* in MS women than in healthy women. We also we found significantly lower transcript levels of *SOCS7* in MS men than in healthy men. There is evidence that the male sex is associated with a poorer clinical outcome in relapse-onset patients (RRMS, SPMS) [34], and that male patients with relapse-onset of MS show a more rapid progression of EDSS scores, with shorter time to SPMS development [35]. This could be supported by our results, which showed that the age range was lower in MS males than females, even though all patients were in the RR phase, indicating that it is possible that MS presents in men at earlier ages than in women. With respect to the EDSS score, we did not find significant differences between MS and healthy women.

The correlation analysis carried out in the present study showed a significant association between *SOCS7* and the EDSS score, which might suggest that MS patients with higher *SOCS7* levels could be in a worse clinical condition, as reflected by a higher score. This contrasted with previous results, from which we inferred that the inhibitory role of SOCS7 on the signaling initiated by proinflammatory cytokines meant that an increase in this mRNA might lead to a reduction of the inflammatory process. Another correlation analysis showed that patients treated with IFN-*β* had high transcript levels *SOCS5*, which were significantly correlated with high EDSS scores. In this regard, it is known that SOCS5 inhibits IFN-*γ*-activated STAT1 and STAT3.

The main limitation of our study is that all patients were in the RR phase. It would be interesting to compare the results of this study with other stages of the disease or with results from a disease relapse to follow the behavior of the transcription genes and the plasmatic cytokine levels. Our results open the door to future studies about the role of SOCS7 in the control of MS.

## 4. Materials and Methods

### 4.1. Patients

A descriptive and transversal study was carried out with 36 MS patients in the relapsing–remitting (RR) phase, recruited from the Neurology Service of the High Specialty Medical Unit of the Mexican Institute of Social Security in Puebla, Mexico, from April 2015 to August 2016, and a group of physically- and neurologically-healthy individuals, matched by age and gender. All subjects gave their informed consent for inclusion before they participated in the study. 

The study was conducted in accordance with the Declaration of Helsinki, and the protocol was approved by Local Committee of Research and Ethics Number 2101 of Mexican Institute of Social Security (27 March 2019) and was registered with number 2015-2101-040. Written informed consent was obtained from all participants 

The gender, age, evolution time, and disease progression (Expanded Disability Status Scale, EDSS) of each subject were recorded. The EDSS score ranges from 1 to 10. Peripheral blood samples were obtained from all participants in EDTA-containing vacuum tubes; they were centrifuged at 2700 rpm for 20 min, the plasm was separated and incubated at −70 °C, and the leukocytes band was extracted and treated with a red blood cell lysis buffer (0.15 M ammonium chloride, 10 mM KHCO3, 0.1 mM EDTA) at 4 °C for 15 min and centrifuged at 1700 rpm for 4 min. The remaining white cells were washed with a buffered saline solution.

### 4.2. RNA Extraction

Total RNA was extracted from leukocytes using Trizol reagent (Invitrogen, Carlsbad, CA, USA), and treated with DNAse I (Fermentas, Glen Burnie, MD, USA). RNA concentration and quality were determined by spectrophotometry and electrophoresis.

### 4.3. Quantitative Real-Time Reverse Transcription-PCR

The method for the quantification of SOCS5 and SOCS7 transcripts has been described before [23,36]. The resulting values were analyzed using the comparative method 2^−ΔΔCT^ (fold difference).

### 4.4. Cytokine Quantification

The plasmatic concentrations of IFN-*γ*, IL17, and IL16 were determined using ELISA MAX Deluxe kits (BioLegend, San Diego, CA, USA). Immunoassays were carried out according to the manufacturer´s instructions. Plasmatic samples had been previously thawed. For each cytokine, a curve with serial dilutions was built from a known standard. All reactions were performed in duplicate.

### 4.5. Statistical Analysis

The Statgraphics Plus (4.1) software (The plains, VA, USA) was used for the analysis of data. Regarding the cytokine plasmatic levels, the analysis was based on the median values. The differences between the two groups were assessed using the Mann–Whitney U test, while the Kruskal–Wallis test was used to identify differences between more than two groups. Values of *p* < 0.05 were considered statistically significant. The Spearman’s correlation coefficient was calculated between transcript levels, cytokine concentrations, and clinical variables.

## 5. Conclusions

An increase in the transcript levels of SOCS7 was found in patients treated with GA compared to patients with IFN-*β*. These patients also showed a reduction in IFN-*γ* levels. The increase in IFN-*γ* in MS patients compared with the control group could help explain the immunopathology of MS, while the regulation of IFN-*γ* by SOCS5 could help explain its pathophysiology. The differences in socs transcript levels according to gender could be important in explaining the prevalence of the disease.

## Figures and Tables

**Figure 1 ijms-21-00218-f001:**
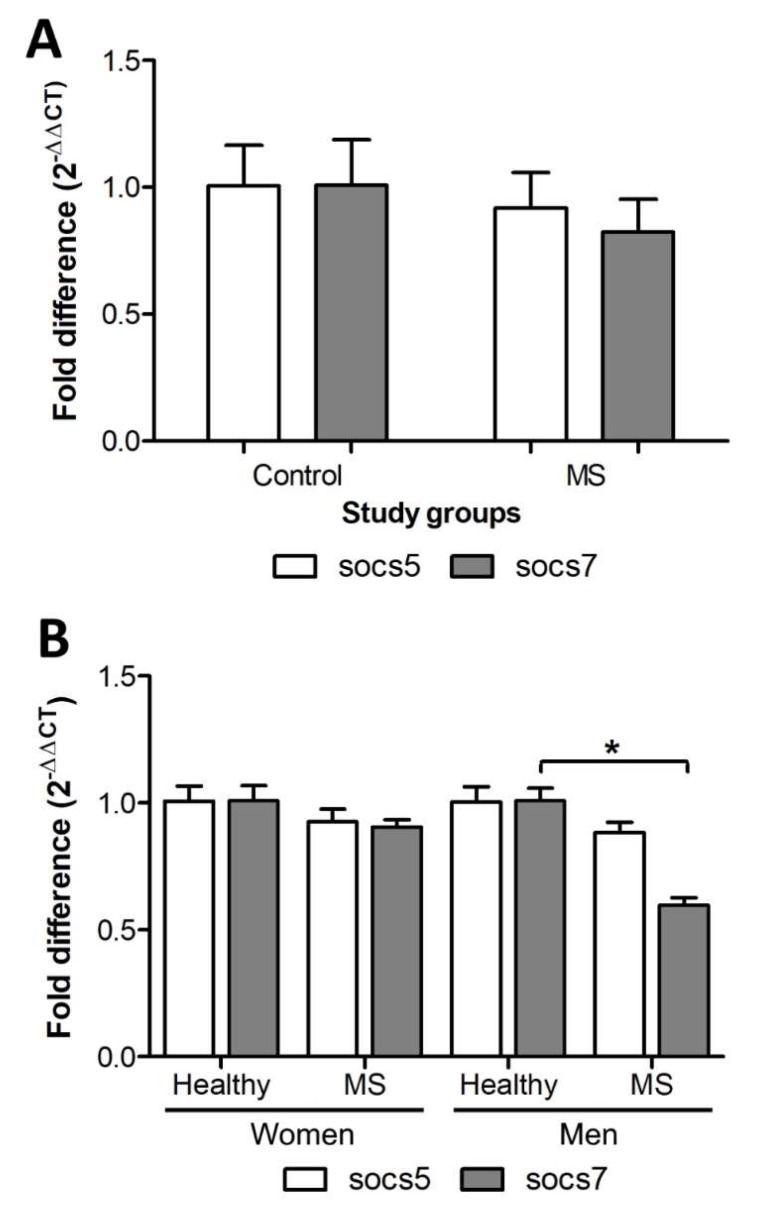
*SOCS5* and *SOC7* transcript levels were quantified by real time RT-PCR, and the fold difference (2^−DDCT^) between the study groups was calculated. Statistical differences were assessed using the Mann Whitney U test, (*) means *p* < 0.05. (**A**) Transcription level in MS patients and the control group. (**B**) Transcription level by gender.

**Figure 2 ijms-21-00218-f002:**
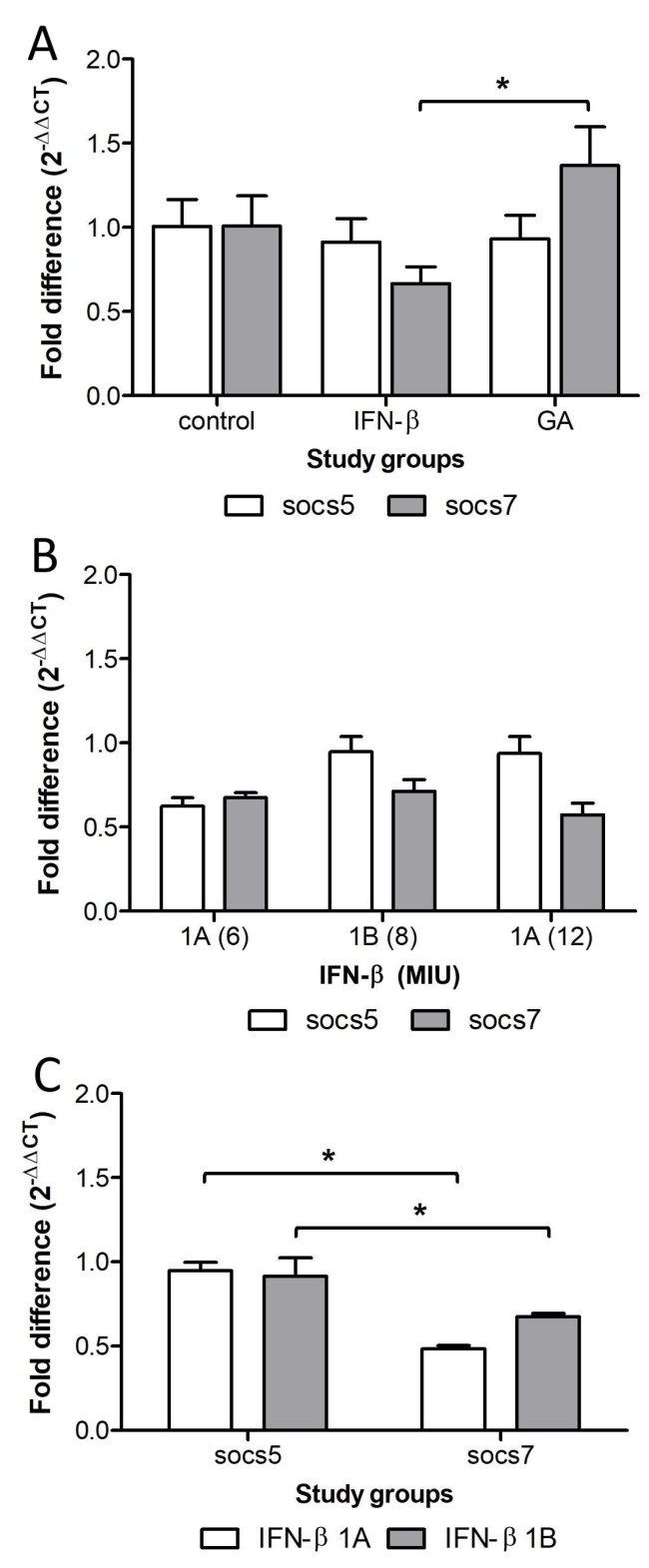
*SOCS5* and *SOCS7* transcript levels in patients under treatment for MS. Transcript levels were quantified by real time RT-PCR, and the fold difference (2^−DDCT^) between the study groups was calculated. Statistical differences were evaluated using the Mann Whitney U and Kruskal Wallis tests, (*) means *p* < 0.05 (**A**) In MS patients treated with IFN-*β* or GA. (**B**) According to the treatment scheme of IFN-*β*: 6, 8 or 12 MIU. (**C**) According to the type of IFN-*β*: 1A or 1B. MIU: Millions of international units. GA: glatiramer acetate.

**Figure 3 ijms-21-00218-f003:**
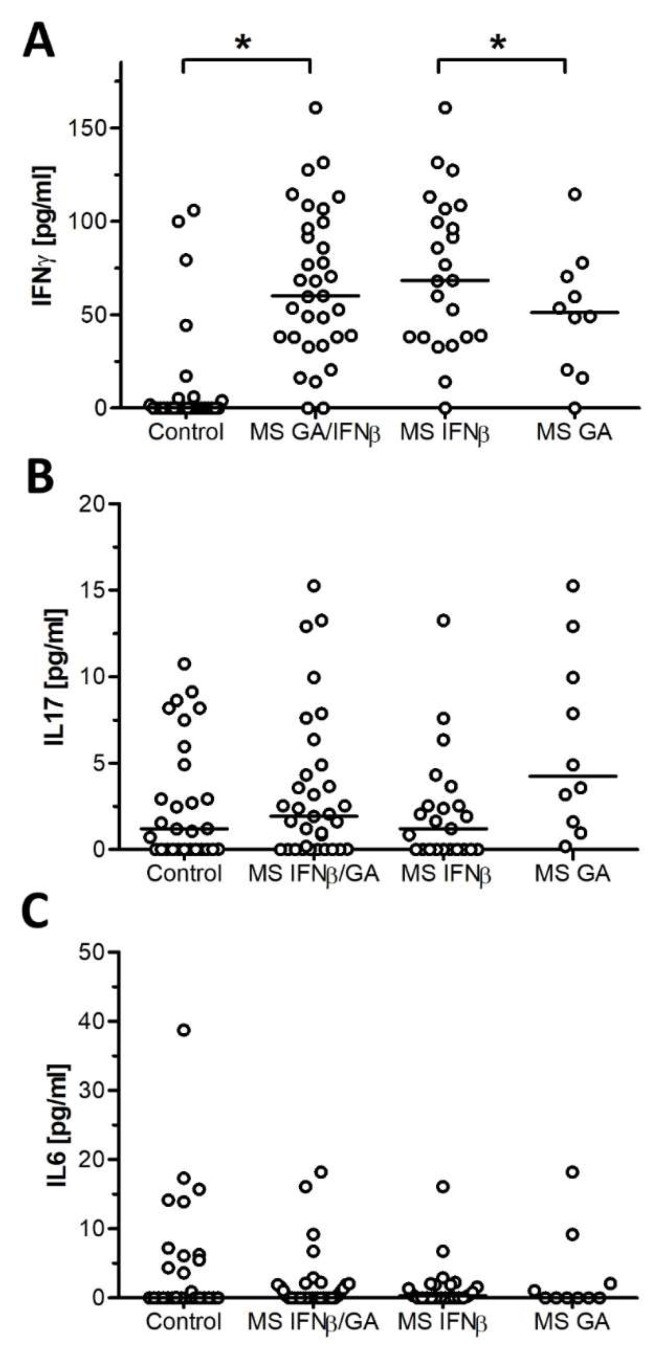
Cytokine concentrations in MS patients and healthy subjects. Cytokines were determined using ELISA MAX Deluxe kits (BioLegend). The results show the values of all subjects belonging to each group. The line represents the median of each cytokine in pg/mL. Significant differences were assessed using the Mann Whitney U test, (*) means *p* < 0.05. (**A**) IFN-*γ*, (**B**) IL17, (**C**) IL6.

**Figure 4 ijms-21-00218-f004:**
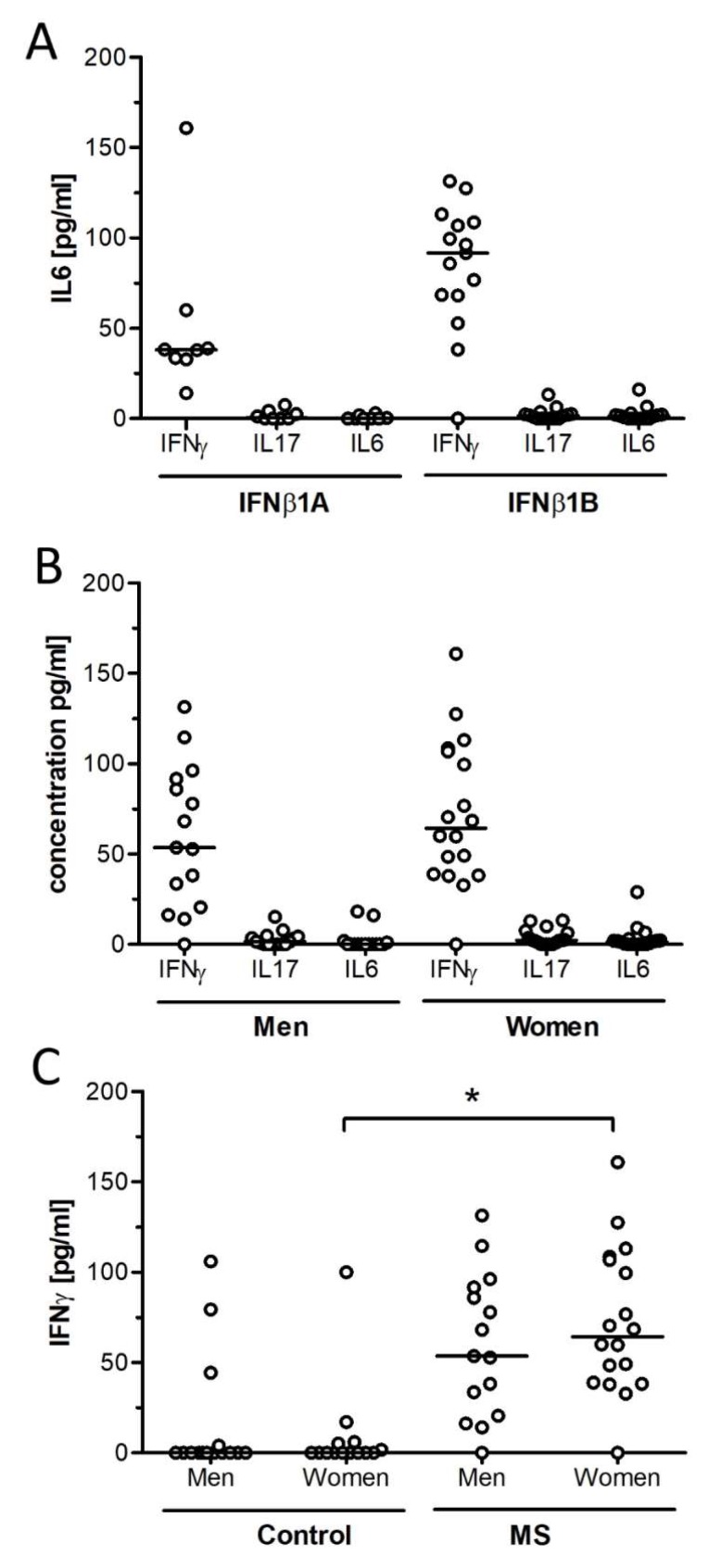
Cytokine concentrations in MS patients. Cytokines were determined using ELISA MAX Deluxe kits (BioLegend). The results show the values of all subjects belonging to each group. The line represents the median of each cytokine in pg/mL. Significant differences were assessed using the Mann Whitney U test, (*) means *p* < 0.05. According to (**A**) IFN-*β* type: 1A and 1B. (**B**) Gender of MS patients. (**C**) Gender of the control and MS patients.

**Figure 5 ijms-21-00218-f005:**
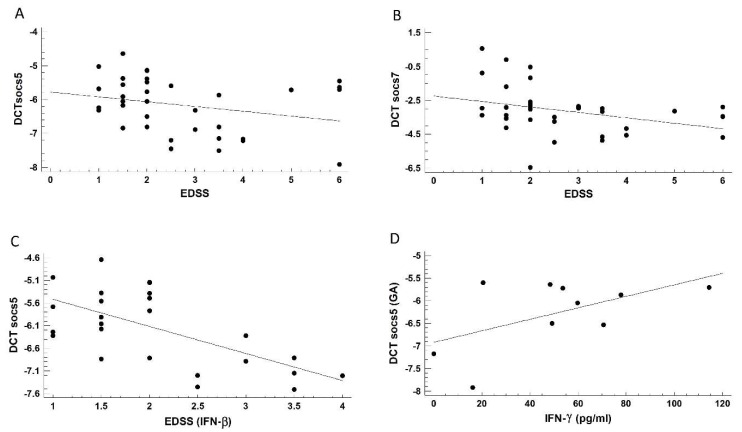
Correlation analysis between socs, EDSS score, and cytokine concentrations. The correlation was calculated using Spearman´s correlation coefficient. (**A**) and (**B**) Correlation between *SOCS5* or *SOCS7* and EDSS score. (**C**) In patients treated with IFN-*β*, a correlation was found between SOCS5 and EDSS. (**D**) In patients treated with GA, there was a correlation between *SOCS5* and IFN-*γ* concentration. EDSS: expanded disability status scale.

**Table 1 ijms-21-00218-t001:** Demographic and clinical characteristics, SOCS5 and SOCS7 transcript levels and cytokine concentrations of MS patients and healthy control individuals.

Variables		Study Groups		Treatment Type		IFN-*β* Types	
		Control (*n* = 29)	MS (*n* = 36)	IFN-*β* (*n* = 25)	GA (*n* = 11)	IFN-*β* 1A (*n* = 10)	IFN-*β* 1B (*n* = 15)
**Age (Years)**		38.5 ± 11.5	36.8 ± 8.7	36.9 ± 9.6	36.5 ± 6.3	39.5 ± 12.7	35.5 ± 7.7
**Gender (Female/Male)**		13/16	21/15	16/9	5/6	7/3	9/6
**EDSS Value**		0	2.6 ± 1.36	2.08 ± 0.88	4.3 ± 1.7	1.85 ± 0.74	2.2 ± 0.96
**Functional Systems Evaluated by EDSS**	**PF *n* (%)**	_	23 (69.7)	14 (60.9)	9 (90)	3 (37.5)	11 (73.3)
	**SF *n* (%)**	_	24 (72.7)	17 (73.9)	7 (70)	5 (62.5)	12 (80)
	**BBF *n* (%)**	_	20 (60.6)	13 (56.5)	7 (70)	6 (75)	7 (46.6)
	**CCF *n* (%)**	_	22 (66.7)	14 (60.9)	8 (80)	4 (50)	10 (66.6)
	**CF *n* (%)**	_	10 (30.3)	7 (30.4)	3 (30)	1 (12.5)	6 (40)
	**BSF *n* (%)**	_	5 (15.2)	2 (8.7)	3 (30)	0	2 (13.3)
	**VF *n* (%)**	_	10 (33.3)	6 (26.1)	5 (50)	1 (12.5)	5 (33.3)
**Evolution of the Disease (Years)**		_	6.5 ± 4.1	6.1 ± 4.6	7.4 ± 3.3	7.3 ± 6.4	5.5 ± 3.4
**Cytokines** **(pg/mL)**	**IFN-*γ***	0 (0–106)	60 (0–160)	68 (0–160)	51 (0–114)	38 (14–160)	91 (0–131)
	**IL-17**	1.2 (0–15)	1.9 (0–15)	1.2 (0–14)	4 (0.2–15)	0.6 (0–7.6)	1.7 (0–13)
	**IL-6**	0 (0–39)	0 (0–18)	0.3 (0–16)	0 (0–18)	0 (0–2)	1.4 (0–16)
**Genes Expression** **SOCS**	**SOCS5**	1.06 ± 0.16	0.918 ± 0.14	0.912 ± 0.14	0.931 ± 0.1	0.913 ± 0.11	0.947 ± 0.05
	**SOCS7**	1.008 ± 018	0.823 ± 0.13	0.65 ± 0.10	1.368 ± 0.2	0.484 ± 0.02	0.674 ± 0.02

GA, glatiramer acetate; EDSS, Expanded Disability Status Scale; PF, pyramidal function; SF, sensory function; BBF, bladder bowel function; CCF, cerebral cognitive function; CF, cerebellar function; BSF, brain stem function; VF: visual function. In the control group (“_”) means that EDSS value is considered zero, because they are healthy individuals.

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
