# Peer review of "Differential Transcription of SOCS5 and SOCS7 in Multiple Sclerosis Patients Treated with Interferon Beta or Glatiramer Acetate"

_ijms, 2019, doi:10.3390/ijms21010218_

Round 1
Reviewer 1 Report
The study is well written and provides new insights and steps forward into understanding the pathology of MS including cytokine and soc transcription levels according to levels in healthy ones and, in addition, during the treatment of MS with GA or IFN-β.
Author Response
Dear Reviewer:
Thanks for your comments about our manuscript.
Reviewer 2 Report
The manuscript by Morales et al. describes that in multiple sclerosis (MS) patients treated with either interferon beta (IFN-β) or glatiramer acetate (GA) there is a differential expression of SOCS5 and SOCS7. They further speculate that this could partially explain the mechanism of action of GA and could help in elucidating the immunopathology of MS. However, there are a few concerns regarding the manuscript:
The data presented in the abstract is confusing and hard to follow. The introduction focuses on the types of MS and emphasizes on different therapies that are either in clinical trials or used in clinics for MS. In addition, the authors explain the importance of B cells in MS. These facts seem to be irrelevant to the current work. The authors should rather give more emphasis on the cytokines and the known functions of SOCS5 and SOCS7 in general and in MS. “Twenty-five patients were treated with 6, 8 or 12 MIU (Millions of International Units) of IFN-γ, once or three times a week”. What does this mean? Which patients were treated once and who were treated thrice a week? In table 1 what do AG means? Is it a typo error? The authors should mention the demographics of male and female MS patients as they show a decrease of SOCS7 in males and not in females. “The treatment with IFN-β was applied in three doses: 6, 8 and 12 MIU. The first and third doses correspond to IFN-β 1A, and the second dose to IFN-β 1B”. The description, as well as the corresponding figure, is confusing to understand. The authors show the decrease in SOCS7 in MS males and later show an increase of SOCS7 by GA treatment in MS. Does this hold up for MS males as well? “With respect to the type of IFN-β, in MS patients treated with IFN-β 1A, SOCS5 transcription was higher (0.94±0.05) compared to SOCS7 transcription (0.48±0.02). For IFN-β 1B, SOCS5 transcription was higher (0.91±0.11) compared to SOCS7 transcription (0.67±0.02). (Figure 2C)". There is no such correlation or statistics shown in the figure. With respect to cytokine measurement in plasma of healthy and MS patients, the data described doesn’t match the representative figure shown. The authors describe an increase or decrease without showing the stats. What is the level of IL-10 secretion? The discussion has several errors. The authors describe some data as increased or decreased and then mention they were not significantly different. These types of interpretation should be avoided. The facts are over speculated without showing concrete data to support their hypothesis. There are several typos, grammatical errors and complex structuring of sentences. For example, line 103 “SOCS5 and SOCS7 transcripts showed no significantly lower expression (p=0.3) in MS patients”, line 125, etc. Therefore, I recommend the authors have a person whose native language is English proofread the manuscript.Author Response
Dear Reviewer
Thanks for your comments and suggestions, below you find the answers.
The manuscript by Morales et al. describes that in multiple sclerosis (MS) patients treated with either interferon beta (IFN-β) or glatiramer acetate (GA) there is a differential expression of SOCS5 and SOCS7. They further speculate that this could partially explain the mechanism of action of GA and could help in elucidating the immunopathology of MS. However, there are a few concerns regarding the manuscript:
1.The data presented in the abstract is confusing and hard to follow.
Answer: the abstract has been modified to make it clearer.
2.The introduction focuses on the types of MS and emphasizes on different therapies that are either in clinical trials or used in clinics for MS. In addition, the authors explain the importance of B cells in MS. These facts seem to be irrelevant to the current work. The authors should rather give more emphasis on the cytokines and the known functions of SOCS5 and SOCS7 in general and in MS.
Answer: the introduction has been modified, following your suggestions. Information was added about socs5 and socs7, and proinflammatory cytokines IFN-g, IL17 and IL6.
“Twenty-five patients were treated with 6, 8 or 12 MIU (Millions of International Units) of IFN-β, once or three times a week”. What does this mean? Which patients were treated once and who were treated thrice a week?Answer: The treatment scheme was administered in a sequential way, first, IFN-β doses of 6 MIU once a week; if the patients showed no improvement, that is, if they suffered new attacks or they became more severe, the treatment was changed to IFN-β doses of 12 MIU administered three times a week. Again, if the patients didn't show any improvement (considering the same criteria), the treatment was changed to IFN-β doses of 8 MIU administered every third day. Glatiramer acetate was administered as the last alternative following the same criteria. When we obtained samples of MS patients, 25 MS patients had treatment with IFN-β, of which; 2 patients treated with IFN- β 6 MIU, 8 patients with IFN- β 12 MIU and 15 patients treated with IFN- β 8 MIU.
To sum up, 6 MIU of IFN-b were administered once a week, 12 MIU of IFN- β were administered three times a week, and 8 MIU of IFN-b were administered every third day. This information is found on page 3, lines 95-97, and page 6, line 154 to 158
In table 1 what do AG means? Is it a typo error?Answer: It was a typo error, GA (glatiramer acetate) it was corrected in the table, page 4
5.The authors should mention the demographics of male and female MS patients as they show a decrease of SOCS7 in males and not in females.
Answer: The MS group included 15 males and 21 females whose average age was: 37.4±8.5 (24-55) and 38.09 ±9.6 (23-61) respectively. Page 3 line 87-89
“The treatment with IFN-β was applied in three doses: 6, 8 and 12 MIU. The first and third doses correspond to IFN-β 1A, and the second dose to IFN-β 1B”. The description, as well as the corresponding figure, is confusing to understand.Answer: The figure 2B and the information have been modified. Pages 7 and 6, lines 164 to 167, respectively.
The authors show the decrease in SOCS7 in MS males and later show an increase of SOCS7 by GA treatment in MS. Does this hold up for MS males as well?Answer: We calculated the transcript level of SOCS7 in MS males treated with GA, and we found that the transcription was 1.016; for healthy males, it was 1.008. We inferred that treatment with GA might restore the socs7 levels in MS males, elevating them to the level of the control group.Pages 5 an 6, lines 149-153.
“With respect to the type of IFN-β, in MS patients treated with IFN-β 1A, SOCS5 transcription was higher (0.94±0.05) compared to SOCS7 transcription (0.48±0.02). For IFN-β 1B, SOCS5 transcription was higher (0.91±0.11) compared to SOCS7 transcription (0.67±0.02). (Figure 2C)”. There is no such correlation or statistics shown in the figure.Answer: Yes, it was an omission error; we have already added the stats in Figure 2C and in the text. Page 6, line 166
9.With respect to cytokine measurement in plasma of healthy and MS patients, the data described doesn’t match the representative figure shown. The authors describe an increase or decrease without showing the stats.
Answer: Yes, it was an omission error; we already added the stats in paragraph 2.4 (Quantification of cytokines), page 8.
10.What is the level of IL-10 secretion?
Answer: We didn´t quantified IL10
The discussion has several errors. The authors describe some data as increased or decreased and then mention they were not significantly different. These types of interpretation should be avoided. The facts are over speculated without showing concrete data to support their hypothesis.Answer: The discussion has been modified; results without stats were removed, and the information was corrected.
12.There are several typos, grammatical errors and complex structuring of sentences. For example, line 103 “SOCS5 and SOCS7 transcripts showed no significantly lower expression (p=0.3) in MS patients”, line 125, etc. Therefore, I recommend the authors have a person whose native language is English proofread the manuscript.
Answer: The text has been proofread by a professional editor. Sentence in lines 103 and 125, (now ….) were corrected.
Round 2
Reviewer 2 Report
I thank the authors for responding to my queries